# Combination of Tipifarnib and Sunitinib Overcomes Renal Cell Carcinoma Resistance to Tyrosine Kinase Inhibitors via Tumor-Derived Exosome and T Cell Modulation

**DOI:** 10.3390/cancers14040903

**Published:** 2022-02-11

**Authors:** Jacob W. Greenberg, Hogyoung Kim, Miae Ahn, Ahmed A. Moustafa, He Zhou, Pedro C. Barata, A. Hamid Boulares, Asim B. Abdel-Mageed, Louis S. Krane

**Affiliations:** 1Department of Urology, Tulane University School of Medicine, New Orleans, LA 70012, USA; jgreenberg@tulane.edu (J.W.G.); mahn@tulane.edu (M.A.); amoustafa@tulane.edu (A.A.M.); amageed@tulane.edu (A.B.A.-M.); 2Department of Zoology and Entomology, Faculty of Science, Helwan University, Cairo 11790, Egypt; 3Department of Physiology & Internal Medicine, Section of Nephrology, Tulane University School of Medicine, New Orleans, LA 70012, USA; hzhou5@tulane.edu; 4Department of Internal Medicine, Section of Hematology/Oncology, Tulane University School of Medicine, New Orleans, LA 70012, USA; pbarata@tulane.edu; 5The Stanley Scott Cancer Center, Ouisiana Cancer Research Center, School of Medicine, Louisiana State University Health Sciences Center, New Orleans, LA 70112, USA; hboulr@lsuhsc.edu; 6Department of Pharmacology, Tulane University School of Medicine, New Orleans, LA 70012, USA; 7Tulane Cancer Center, Tulane University School of Medicine, New Orleans, LA 70112, USA; 8Southeastern Louisiana Veterans Health Care System, New Orleans, LA 70119, USA

**Keywords:** renal cell carcinoma, exosomes, extracellular vesicles, PD-L1, tipifarnib, ERK pathway, sunitinib, resistance, TKI resistance, cell culture

## Abstract

**Simple Summary:**

Metastatic renal cell carcinoma continues to have a poor prognosis. Chemotherapies and immuno-oncologic therapies have garnered increasing importance in cancer therapy, with improvements in patient care and survival. However, a large proportion of patients present with tumors resistant to these treatments. Exosomes are small extracellular vesicles secreted by all nucleated cells that have proven to be key actors in this resistance. Exosomes carry bioactive oncogenic cargos that reprogram target cells to promote tumor growth, migration, metastasis, immune evasion, and chemotherapy resistance. Tipifarnib, in combination with standard therapy, decreased tumor growth in the setting of chemotherapeutic resistance through an exosome-mediated mechanism. After using a qNANO IZON system to compare tumor-derived exosomes collected from untreated and tipifarnib-treated cells, all cancerous cell lines displayed a reduction of vesicle concentration. Tipifarnib also directly inhibited PD-L1 protein expression in chemo-sensitive cell lines and resistant cell lines.

**Abstract:**

Background: Tyrosine kinase inhibitors (TKI) were initially demonstrated as an efficacious treatment for renal cell carcinoma (RCC). However, after a median treatment length of 14 months, a vast majority of patients develop resistance. This study analyzed a combination therapy of tipifarnib (Tipi) + sunitinib that targeted exosome-conferred drug resistance. Methods: 786-O, 786-O-SR (sunitinib resistant), A498, A498-SR, Caki-2, Caki-2-SR, and 293T cells were cultured. Exosomes were collected using differential ultracentrifugation. Cell proliferation, Jurkat T cell immune assay, and immunoblot analysis were used for downstream analysis. Results: SR exosomes treatment displayed a cytotoxic effect on immune cells. This cytotoxic effect was associated with increased expression of PD-L1 on SR exosomes when compared to sunitinib-sensitive (SS) exosomes. Additionally, Tipi treatment downregulated PD-L1 expression on exosomes derived from SR cell lines. Tipi’s ability to downregulate PD-L1 in exosomes has a significant application within patients. Exosomes collected from patients with RCC showed increased PD-L1 expression over subjects without RCC. Next, exosome concentrations were then compared after Tipi treatment, with all SS cell lines displaying an even greater reduction. On immunoblot assay, 293T cells showed a dose-dependent increase in Alix with no change in either nSMase or Rab27a. Conversely, all the SS and SR cell lines displayed a decrease in all three markers. After a cell proliferation employed a 48-h treatment on all SS and SR cell lines, the drug combination displayed synergistic ability to decrease tumor growth. Conclusions: Tipifarnib attenuates both the exosome endosomal sorting complex required for endosomal sorting complex required for transport (ESCRT)-dependent and ESCRT-independent pathways, thereby blocking exosome biogenesis and secretion as well as downregulating PD-L1 on SS and SR cells.

## 1. Introduction

Renal cell carcinoma remains a commonly diagnosed malignancy. Every year in the United States alone, more than 76,000 individuals will be diagnosed with this disease, and over 13,000 people will die [1]. While prognosis remains excellent for localized lesions, survival decreases dramatically in patients with metastatic deposits. It is suspected that of all patients diagnosed with RCC, 25 to 30% initially present with metastatic lesions [1]. Dual first-line therapy for treatment-naive patients with advanced RCC is with anti-programmed death-ligand (PD-L1) immunotherapy and tyrosine kinase inhibitors (TKI) [2]. The TKI utilized in this study was sunitinib. Sunitinib’s primary mechanism of action blocks the vascular endothelial growth factor receptor, thereby inhibiting downstream signaling of PI3K, AKT, and mTOR, causing downregulation of tumor angiogenesis and proliferation [2].

While the disease is still localized, surgical extirpation is appropriate; however, if left untreated, premetastatic seeding, with communication between sites with distinct genomic signatures, allows for disease progression [3]. Current therapy targets immunosuppression and cell proliferation, however, mechanisms to limit cellular communication may provide a novel pathway for targeting the premetastatic niche comprised of tumor-derived secreted factors (TDSFs), extracellular vesicles (EVs) specifically exosomes, bone marrow-derived cells (BMDCs), suppressive-immune cells, and host stromal cells [4]. The most common mechanism of cellular communication which cancer utilizes, tumor-derived exosomes, are prime targets given an exosome’s functional composition. Exosomes range from 50 to 150 nm in size are composed of a lipid bilayer that encapsulates bioactive molecules such as proteins, lipids, metabolites, functional RNA, and DNA [5,6,7]. More specifically, the oncogenic cargo of these tumor-derived exosomes can be functionally or passively integrated into distant metastatic tumors, thus leading to novel drug resistance, thereby facilitating the tumor’s growth within extrarenal microenvironments [8,9]. Exosomes are differentiated from other extracellular vesicles such as microvesicles (150–1000 nm) and apoptotic bodies (150–5000 nm) based on size and surface or biogenesis markers. Exosomes can be further differentiated from other EVs by the mechanism of biogenesis. Exosomes go through a process of inward budding from the cell surface to form the early endosome. These early endosomes later mature into multivesicular bodies which fuse with the cellular membrane allowing for the release of exosomes to the extracellular space. Microvesicles on the other hand are simply formed through membrane budding [10,11]. Our lab previously identified that tipifarnib has the ability to target exosome biogenesis, the exact mechanism is the topic of this study. Tipifarnib has been established as a farnesyltransferase inhibitor targeting HRAS. Farnesyltransferase is utilized to anchor RAS to the cytosolic side of the cellular membrane. KRAS and NRAS can be processed by both farnesyltransferase and geranyltransferase, while HRAS can only be anchored through the former. By preventing membrane binding, all downstream HRAS signalizing is disturbed [12].

Additionally, tumor-specific exosomes are also prime anti-tumor targets since they have been shown to play a vital role in malignant transformation and immune suppression [13,14], specifically causing NK-cell dysfunction, inhibiting antigen-presenting cells, blocking T cell activation, and enhancing T cell apoptosis to block the adaptive immune response [15,16,17,18]. Recently, researchers confirmed that cancer-derived exosomes carry PD-L1, which functions as an immune-surveillance modulator, inhibiting lymphocytes activation by binding to PD-1 on T cells [3,12]. Currently, immunotherapy with PD-1/PD-L1 inhibitors has made the translational jump to clinical medicine. First-line immunotherapy blocks PD-L1 from interacting with PD-1, allowing for the activation of immune cells that were previously inhibited by RCC. However, patients treated with PD-1/PD-L1 inhibitors have significant individual differences in expression, T cell receptors’ makeup, and response to therapy [19,20]. 

This study aimed to determine if tipifarnib could modulate renal cell carcinoma derived exosome and PD-L1 expression. This study investigated the pathways associated with exosomal biogenesis and secretion to determine if tipifarnib can modulate renal cell carcinoma-derived exosome expression and augment exosome conferred sunitinib resistance. Results demonstrated that newer agents inhibited tumor/immune-system interactions through downregulating tumor-derived exosomes. 

## 2. Materials and Methods

### 2.1. Sample Collection

Prior to all experiments, this study’s research protocol was approved by the Tulane Institutional Review Board (IRB). IRB #2017-483-TUHSC was granted by the Tulane University Health Sciences Center (TUHSC), Human Research Protection Office and Institutional Review Board. Full consent was obtained from each patient for use of their biological samples in this study. The study protocol was conducted in accordance with current guidelines and federal/local regulations. The normal plasma samples (the control group) were purchased from BioIVT elevating science (Westbury, NY, USA) [21]. 

### 2.2. Materials

Cell-culture mediums RPMI 1640, penicillin/streptomycin solution, and fetal bovine serum (FBS) were all purchased from Invitrogen (Camarillo, CA, USA). Tipifarnib and sunitinib were obtained from Selleckchem (Houston, TX, USA). All additional chemicals used in this study were obtained from Sigma (St. Louis, MO, USA) [21]. 

### 2.3. Cell Culture

Cell lines 786-0, Caki-2, A498, 293T, and Jurkat were purchased from ATCC (Manassas, VA, USA). Sunitinib-sensitive (SS) cell lines were cultured in RPMI 1640 medium supplemented with 10% fetal bovine serum, 2 mM L-glutamine, and 1% penicillin/streptomycin (P/S). Routine maintenance was conducted on each cell line by assessing a monolayer culture at 37 °C, in an incubator with 5% CO_2_ and 95% air. Sunitinib-resistance (SR) was induced in these cell lines (786-0-SR, Caki-2-SR, and A498-SR) by treating sensitive cells with increasing dosages of sunitinib for six months. Once SR was confirmed at high dosages, cultures were maintained by adding 5 μM sunitinib to cell media for more than 10 passages as previously described [21]. Full images for Jurkat T cells detected with triple staining can be found in Appendix A. Additionally, full image for HRAS mutation status and effect of varying tipifarnib dose on cell viability of our cell lines can be found on Appendix A.

### 2.4. Exosome Isolation and Characterization

Conditioned media (CM) was produced by plating with complete media, and cells were allowed to attach overnight. Following overnight growth, the culture media was swapped for an FBS-exosome-depleted media that contained either a control or experimental drug treatment. Conditioned media (CM) was produced by plating with complete media, and cells were allowed to attach overnight. Following overnight growth, the culture media was swapped for an FBS-exosome-depleted media that contained either a control or experimental drug treatment. In order to create FBS-exosome-depleted media, FBS was diluted with PBS in a ratio of 5:5 (FBS:PBS). These preparations were then subjected to 18-h (overnight) centrifugation at 104,000× *g* (SW28 rotor, Beckman Coulter, Indianapolis, IN, USA). The supernatants were then used to prepare 10% FBS-containing medium that was never in contact with cells. Cells were then allowed to grow for 48 h. Subsequently, media was extracted from the plate, and the supernatant was isolated for exosome isolation as previously described. Finally, exosomes were purified by using differential ultracentrifugation (DU) as described previously [21]. 

Exosome concentrations, size distributions, and diameters of different CM samples from cells treated with Tipi or control vehicle (DMSO) were measured, using tunable resistive pulse sensing (TRPS) technology (qNano IZON system; Izon, Cambridge, MA, USA). The qNano system was calibrated prior to each read by standardizing voltage, stretch, pressure, and baseline current, using IZON proprietary beads named CPC100 (mode diameter: 114 nm, concentration: 1.0 × 10^13^/mL) and CPC70 (mode diameter: 70 nm, concentration: 3.0 × 10^13^/mL). A diluted sample size of 40 μL and NP100 nanopores (for 50–250 nm size range) were used, and data analysis was performed by qNano IZON Control Suite software [22]. A full flowchart of the exosome isolation steps used can be found in Appendix A. Furthermore, the effects of tipifarnib on exosome diameter excreted by sunitinib sensitive and sunitinib resistant cells can be found in Appendix A.

### 2.5. Cell Viability and the Cytotoxicity 

RCC cells were seeded in 96-well culture plates (Corning) at a density of 1 × 10^4^ cells/well and incubated for 48 h (for RCC) or 72 h (for Jurkat T cell) at 37 °C, 5% CO_2_ in a humidified CO_2_ incubator. Thereafter, RCC cells were treated with various concentrations of tipifarnib and sunitinib dissolved in DMSO for 48 h at 37 °C, 5% CO_2_ in a humidified CO_2_ incubator [23]. Cell viability was measured by the MTT (methylthiazolyldiphenyl-tetrazolium bromide) (Sigma-Aldrich, St. Louis, MO, USA) cell cytotoxicity assay according to the manufacturer’s protocol as we described [21]. Cell viability was determined by incubating the treated cells with MTT solution (5 mg/mL) for 1 h at 37 °C. DMSO was then added to the wells, and the optical density (O.D.) of formazan crystals solubilized in DMSO was measured at 570 nm by using a µQuant spectrophotometric plate reader from Bio-Tek (Seattle, WA, USA) [21]. 

To determine cell viability and cytotoxicity, cells were treated with exosomes at a single dose of 40 µg. At the onset of the experiments, the Jurkat T cell line was seeded at an optimal density (1 × 10^4^ cells per well) in a 96-well plate with or without exosomes for 72 h. The viability of Jurkat T cells was determined by using the WST-8 cell viability assay kit (Dojindo Molecular Technologies, Inc., Rockville, MD, USA). After the determined time point, the medium containing the chemical was removed, and WST-8 solution dissolved in the medium was added to each well before incubating the cells for 1 h at 37 °C, 5% CO_2_ in a humidified CO_2_ incubator. Absorbance was subsequently measured at 450 nm, using a µQuant spectrophotometric plate reader from Bio-Tek as described [23].

### 2.6. Transfection, Immunofluorescence, and Western Blot Analyses

The pEGFP-N1/PD-L1 plasmid was purchased from Addgene (Watertown, MA, USA). A498 cells were transiently transfected with the pEGFP-N1/PD-L1 plasmid, using Lipofectamine 3000, according to the manufacturer’s instructions (Invitrogen). Cells were visualized by using a Nikon fluorescence microscope (magnification, ×20).

First, we prepared 8-microwell chamber slides for immunofluorescence experiments for quantifying apoptosis. We added 150 µL of human FNC coating mix (Athena Enzyme Systems, Baltimore, MD, USA) per well to an 8-microwell chamber slide and incubated it for 30 min–1 h at 37 °C. 

Human FNC coating mix was aspirated, using a 200 μL automatic pipette, and each well with 200 µL of PBS was washed and gently shaken for 2 min. After repeating this wash, Jurkat T cells were incubated with 40 ug of exosomes for 24, 48, and 72 h, then incubated with Hoechst 33342 (ThermoFisher Scientific, Waltham, MA, USA), Cell Mask Green plasma membrane stain (Invitrogen) and PI (ThermoFisher scientific) for 15 min at RT in a dark room. Hoechst 33342, Cell Mask, and PI fluorescence were immediately observed under a Nikon fluorescence microscope.

Total proteins were isolated, using standard protocols as previously described [24]. Protein extracts were utilized for western blot analyses. Antibodies, employed in this assay, targeted Alix, pERK, ERK, cyclin D, cleaved caspase-3, caspase-3, cleaved caspase-9, cleaved PAPR-1, and PARP-1. After treating blots with primary antibody, the appropriate secondary was bought from Jackson ImmunoResearch, Inc. Luminescent reagents were then added to illicit a signal as described [21]. The Image Quant Las 300 from GE healthcare was used to image each blot and ImageJ from the NIH was then used to quantify. Additional primary antibodies included Mutation Ras antibody kit from Cell Signaling, Actin (Santa Cruz), CD81 (Santa Cruz), Flotillin-1 (Santa Cruz), nSMase (Abcam), PD-L1 (Abcam, Waltham, MA, USA), and Rab27a (Proteintech, Rosemont, IL, USA). Fold change difference of exosome biogenesis and secretion markers after tipifarnib treatment on our cell lines can be found in Appendix A. Further full blot images can be found in Appendix A.

### 2.7. Colony Forming Units (CFU) Assay

A CFU assay was employed to assess the long-term effects of treatment. Furthermore, 500 cells per dish were grown on a 6-well plate using RPMI 1640 media supplemented with 2% FBS. All CFU assays were completed in three replicates. After 48 h of incubation, study drug was added to each well and replaced on a weekly basis. Culture wells were stained after two weeks of incubation. Cells were first fixed to the plate using 100% ethanol, 0.2% crystal violate, and 20% methanol. ImageJ from the NIH was then used to count the colony forming units (CFU) and later compared to drug untreated wells as previously described [21]. 

### 2.8. Analysis of the Correlation between Exosome Biogenesis and Secretion Markers and RCC

Patients’ clinical, pathological, and laboratory reports were collected from the TCGA database (TCGA (https://tcga-data.nci.nih.gov/tcga/ (accessed on 25 January 2021)) and correlated to different outcome variables, including overall survival and T stage. The corresponding mRNA data were also collected, using cBioportal (http://www.cbioportal.org/publicportal/ (accessed on 25 January 2021)). Using R computational language (Version 3.5.3), Kaplan–Meier OS (overall survival) probability plots were generated, using “survival” and “survminer” packages. The Kruskal–Wallis test was used to compare various non-parametric continuous variables between T stages. Exosome concentration data are presented as means ± S.E.M. with more than three independent experiments performed in triplicate. For western blots, a case representative experiment is depicted in the figures section. Multiple comparisons among groups in western blot were performed using ANOVA with Bonferroni correction in R. All tests were two-tailed, using a significance level of 0.05 as previously described [25].

### 2.9. Annexin V-FITC/PI Double-Staining Assay

Apoptotic cells were quantified by flow cytometry, using an eBioscience Annexin V Apoptosis Detection Kit FITC (ThermoFisher Scientific). Jurkat cells were seeded in 12-well plates, then treated with PBS (vehicle) or 40 µg normal-derived exosomes and 40 µg RCC-derived exosomes for 24 h at 37 °C. Jurkat T cells were isolated and phosphate-buffered saline was then used to wash them twice. Cells were then stained, using the eBioscience Annexin V Apoptosis Detection Kit FITC (Invitrogen) outlined in the manufacturer’s instructions. All stained cells were processed by MACSQuant analyzer 10 and Flowlogic Analysis Software.

### 2.10. Real Time-PCR 

An RT-PCR was conducted in this study to assess changes in mRNA expression after treatment. After growing cells, RNA was extracted using a kit from QIAGEN called the RNasey Plus Micro Kit (Valencia, CA, USA). Total RNA was then used to make cDNA libraries. This was accomplished using a kit from Bio-Rad called the iScriptTM cDNA Synthesis kit. Primer sequences employed can be found in Appendix A. Primers were ordered from Integrated DNA Technologies. The top-down PCR parameters included 30 s at 95 °C, 30 s at 60 °C, and 30 s at 72 °C. This study used 45 denaturation, annealing, and extension cycles to allow for sufficient product. Ct values were then compared between groups after being normalized to GAPDH. Samples were run in triplicates to reduce sample variation. A list of primer sequences used in this study can be found in in Appendix A.

### 2.11. Statistical Analysis

Using GraphPad Prism, means were compared for samples with ≥3 independent variables, all performed in triplicate. When comparing variables between >2 groups, an ANOVA with Bonferroni correction was employed. Significance for all tests was set to 0.05, as previously described [21,24].

## 3. Results

### 3.1. SR RCC Exosome’s Role in Suppressing the Immune System

First line therapy for metastatic renal cell carcinoma and gastrointestinal stromal tumors includes sunitinib which is a protein tyrosine kinase inhibitor targeting VEGF-R, c-kit, and PDGF-R. While sunitinib has shown clinical application for the treatment of advanced renal cell carcinoma (RCC), the literature has identified drug resistance as the main barrier to improved long-term outcomes. Exosomes exchanging information between cells, locally and distantly, has been linked to drug resistance [12,13,14]. More specifically, tumor-derived exosomes contribute to the formation of the premetastatic microenvironment, tumor growth and progression, immune escape, angiogenesis, anti-apoptotic signaling, drug-resistance, and so on [5,6]. Since exosomes biogenesis/secretion inhibitors were efficacious on prostate and RCC cell lines, as previously described [21,26], tumorigenic properties of the exosomes secreted by SR RCC cells were investigated to highlight the importance of targeted therapy. To assess the role of SR exosomes within the tumor microenvironment, the immortalized human T lymphocyte cell line, Jurkat, that is commonly used to evaluate in vitro immune system modulation, was employed for cytotoxicity assay. In our study, protein concentrations in exosome preparations were determined using BCA for exosomes and using qNano IZON system for exosomes particle concentration. It has been shown that exosomes show particle-to-protein ratios (particles per microgram of protein) (Appendix A). Exosomes were extracted from conditioned media from each cell line, using ultracentrifugation, and the Jurkat T cells were subsequently segregated into groups of increasing dosages from 0 to 40 µg of exosomes. After 72 h of incubation, cell cytotoxicity was measured, using WST-8 assay. We used exosomes extracted from 293T cells as a negative control. Jurkat T cells displayed no alterations in proliferation after a 72-h incubation of 293T exosomes. Next, we measured the difference between exosomes collected from SS and SR cell lines. Exosomes extracted from all SS and SR cell lines, independently increased cytotoxicity of Jurkat T cells at concentrations of 40 µg. However, when compared to SS exosomes, exosomes from SR cells had an even greater Jurkat T cell toxicity profile (*p* < 0.05). Notably, Caki-2-SR exosomes reduced the proliferation of Jurkat T cells at 20 µg while SS exosomes failed to exhibit a cytotoxic effect, *p* < 0.05 (Figure 1A). To investigate if exosome-induced cytotoxicity of Jurkat cells is linked to the increment of apoptosis signaling markers, we analyzed the hallmarks of apoptosis using western blot analysis. As expected, the treatment of Jurkat cells with 40 µg SR exosomes was associated with the cleavage PARP-1, caspase-3, and caspase-9 over SS exosome dosing (Figure 1B). In short, treatment with SR exosomes significantly induced the apoptosis of Jurkat T cells, compared to SS exosomes. A comparison of exosome protein concentration to partial concentration can be found on Appendix A.

### 3.2. The Level of PD-L1-Containing Exosomes from SS and SR RCC Cells

In recent years, programmed death receptor 1 (PD-1) and programmed death ligand 1 (PD-L1) have attracted much attention [27,28]. PD-1 is mainly expressed on macrophages, activated T cells, and B cells. PD-L1, on the other hand, is highly expressed on tumor tissues, tumor-associated antigen-presenting cells (APCs), and stromal cells [27,28]. In the absence of PD-L1, T cells can recognize and attack tumor cells. However, when PD-1 binds to PD-L1 expressed by the tumor, this interaction induces an inhibitory signal which cascades into the inhibition of T cell activation and lymphocyte apoptosis. Therefore, blocking the PD-1/PD-L1 pathway can enhance the killing effect of T cells and improve the immune response to cancerous cells [29,30]. To ascertain if the PD-1/PD-L1 pathway explained the reasons why SR RCC exosomes displayed a cytotoxic effect on human Jurkat T lymphocytes, PD-L1 protein concentrations between cells and exosomes were first compared, using western blot analysis (Figure 1C,D). Within cells, an increased PD-L1 expression on A498-SR and Caki-2-SR cell lines was observed when compared to SS counterparts. Within exosomes, PD-L1 trended at even higher concentrations in all SR exosomes when compared to their SS counterparts. The assessed exosomes were validated, using the expression of both CD9 and CD81. The presence of GRP94, a marker specific to large EVs, was then evaluated in all cell and exosome lysates. GRP94 was identified across all cell lysates. However, GRP94 was absent in our exosome lysates displaying that our samples contained predominantly exosomes.

### 3.3. Patient PD-L1 Expression and T Cell Modulation after Tumor-Derived Exosome Exposure 

Clinical significance of PD-L1 levels was determined by assessing exosomal expression within patient blood prior to surgical or medical intervention of RCC. Previous data demonstrated that high levels of exosomes was associated with clinically aggressive disease when compared to blood exosome levels across patients with varying clinical stages. Additionally, patient specific- and in vitro cell culture specific-exosome size distribution was found to be comparable at 50–150 nm [21]. Significantly higher levels of exosomal PD-L1 (*p* < 0.0035) were observed in patients with active disease versus subjects without RCC (Figure 2A).

Two recent studies have highlighted the role of exosomal PD-L1 as an important biomarker clinicians can use to risk-stratify patients entering immunotherapy [31,32]. Historically, PD-L1 was thought to be expressed by cancer and would bind to the PD-1 receptor found on T cells to downregulate immune activation, induce apoptosis, and cause anergy [33,34]. To investigate whether patient exosomes induced apoptosis in Jurkat T cells, the Annexin V-FITC/PI double-staining method was utilized. Flow cytometry showed that the positive control induced a marked increase in Jurkat T cell-specific apoptotic bodies after a 24 h exposure to 10 µM VP-16. Exposure of Jurkat T cells to 40 µg normal exosomes, which were collected from patient plasma without RCC, resulted in no discernable difference in apoptosis when compared to the negative control. When exposed to 40 µg of RCC patient-derived exosomes, Jurkat T cell apoptosis increased significantly (Figure 2B,C). Consistent with these results in Figure 1A,B, treatment with patients’ PD-L1 positive exosomes significantly induced the apoptosis of Jurkat T cells compared with non-RCC patient exosomes.

To better understand the function of PD-L1 containing exosomes, we employed an overexpression model transfected with the PD-L1-EGFP vector within A489 cells. A498 cells were visualized EGFP-expressing PD-L1, via fluorescent microscopy (Figure 2D). The presence of exosomes was then validated within our samples through the presence of tetraspanin (CD81), flotillin-1, PD-L1, and GAPDH (Figure 2E) as described previously [21]. The effect of exosomes on Jurkat T cell apoptosis was evaluated, using Hoechst 33342, Cell Mask Green plasma membrane stain, and PI triple fluorescence staining. Signals were detected at weak levels within our vehicle-control cells. On the other hand, high fluorescence densities were visible in response to PD-L1-containing exosomes compared with A498-exosome treatment (Appendix A). To investigate if A498-PD-L1 exosomes could induce apoptosis in Jurkat T cells, the PI-staining method was employed. Apoptosis increased significantly upon exposing Jurkat T cells to 40 µg exosomes collected from the A498-PD-L1 overexpressing cell line. (Figure 2F). In addition, after treatment with 40 µg of PD-L1 overexpressed exosomes from A498-PD-L1 EGFP cells, Jurkat T cells displayed increased expression of activated apoptosis signaling pathway protein isoforms PARP-1 and caspase-3 compared with A498-exosome treatment (Figure 2G). These results indicated that PD-L1-positive exosomes promoted the apoptosis of Jurkat T cells through PD-1/PD-L1 interaction.

### 3.4. Clinical Significance of Alix, nSMase, and RAB27a Expression in RCC

The endosomal sorting complex required for transport (ESCRT)-dependent and ESCRT-independent are two pathways employed by all nucleated cells to produce exosomes [21]. Specifically, Alix is implicated as an ESCRT-dependent exosome biogenesis protein while nSMase is involved in the ESCRT-independent exosome biogenesis pathway. Comprised of two isoforms, Rab27a and Rab27b, Rab27 is a family of small GTPases implicated in the regulation of vesicle trafficking [35]. Rab27a is functional in the trafficking of exosomes within the cell and is utilized by both the ESCRT-dependent and -independent pathways. Recently, Tsuruda et al. showed that RAB27b is a prognostic marker and a novel therapeutic target in SS and SR RCCs [36]. Therefore, inhibition of exosome release may slow the progression of RCC, thus a potential target for cancer treatment.

The three mRNA signatures of interest to this study were RAB27a, nSMase (SMPD2), and Alix (PDCD6IP). mRNA and overall survival data of 534 patients from the TCGA database were analyzed. RAB27a mRNA expression was shown to be increased in T4-stage cancer when compared to T1, T2, and T3 stages. nSMase mRNA expression also showed an increased expression after comparing more advanced staged cancer to T1. Although, a dissimilar pattern was seen after analyzing Alix. Decreased Alix was associated with T3 staging when compared to T1. However, T4 Alix expression was statistically increased when compared to T3. These three markers were also used to delineate overall expression. On Kaplan–Meier analysis, the 5-year overall survival probability for patients with the top 25% (1 standard deviation above the cohort’s mean) of RAB27a, nSMase (SMPD2), or Alix (PDCD6IP) were found to be 56% (95% confidence interval (CI) 47–67%), 49% (CI 40–59%), and 70% (CI 60–80%), respectively. Patients with either a high expression of RAB27a, nSMase, or both, had a lower overall survival at log-rank *p* < 0.05 and *p* < 0.001, respectively. Patients with high Alix expression displayed no change in overall survival when compared to the remainder of the cohort, log rank *p* = 0.1 (Figure 3A). Within this in vitro model, exosome biogenesis and secretion protein expression were dramatically increased in cells with acquired resistance to sunitinib compared to SS cells (Figure 3B). Thus, this data regarding the role of exosomes in malignant carcinoma offers ESCRT-dependent and ESCRT-independent targets for the development of novel diagnostic and therapeutic targets.

### 3.5. Tipifarnib Reduces Exosome Load in SS RCC Cell Lines

Drug repurposing has the unique potential to offer patients a new treatment option without the arduous process of phased trials. A recent study identified ten candidate drugs that target exosomes by employing a quantitative high throughput screening assay [26]. One of them was tipifarnib which inhibits farnesylation of HRAS, disrupting anchoring to the cell membrane and downstream activation. Current evidence supports the clinical safety and efficacy of tipifarnib’s antitumor effects in multiple cancer types [26]. However, the implications of tipifarnib for the treatment of RCC is yet unknown.

To identify the therapeutic potentials of tipifarnib, SS RCC cells were cultured with different concentrations of tipifarnib for 48 h with subsequent MTT assay to assess cell viability. First, a dose-response assay with an incremental increase in drug concentration was used to determine a physiologically ideal non-cytotoxic therapeutic range displaying an effect on exosome biogenesis and secretion (Appendix A). A graphical representation of this study’s ultracentrifugation workflow can be found on Appendix A. First, the qNano system was utilized to assess if exosome concentrations were modulated by tipifarnib treatment. 293T cells were used as the wild-type (WT) control while cancerous cell lines 786-0, A498, and Caki-2 were considered the experimental groups. After incubating cells dosed with 0.5 µM tipifarnib for 48 h, all cell lines displayed a reduction in average particle (exosome) concentrations. 293T displayed a 14.9% reduction after tipifarnib treatment with a significance of *p* < 0.05 when compared to DMSO control. 786-0, A498, and Caki-2 cell lines showed an even greater reduction in exosome concentration at 51.6%, 58.5%, and 67.8% with a *p* < 0.01 (Figure 4A). Moreover, the particle-size distribution ranged from 50 to 250 nm. The average particle size, distribution, and diameter were found to be comparable between treatment samples (Appendix A).

Western blot analysis was used to determine the molecular basis for this reduction in exosome concentration. Three different markers were utilized: RAB27a, Alix, and nSMase. Prior to western blotting, cells were segregated into 5 distinct groups of 48 h incubations: DMSO control, 0.25, 0.5, and 1 µM tipifarnib. Upon blot imaging, 293T cell displayed a dose-dependent increase in Alix while showing no change over dosage groups in either nSMase or RAB27a. In contrast, after increasing dosages of tipifarnib, 786-0, A498, and Caki-2 cell lines showed a functional decrease in all three markers (Figure 4B). Next, we examined if Caki-2 or Caki-2-SR is associated with downregulation of exosomes biogenesis/secretion-related genes. As shown in Appendix A, Alix, nSMase2 and Rab27a transcript levels decreased in response to tipifarnib in Caki-2 or Caki-2-SR, compared to vehicle-treated cells when measured by RT-PCR.

Ras signaling cascade spans from cell surface receptors to nuclear transcription factors and involved mitogen-activated protein kinase (MEK) and extracellular signal-regulated kinase ERK. Mutations within the Ras protein are commonly located at codons 12, 13, and 61. Mutations at these three codons are considered “activating mutations” and allow the abnormal configuration to remain active even in the setting of GTPase activity [12]. In Appendix A, SS- and SR-RCC cell lines do not have mutant HRAS by specific mutant RAS antibodies (G12V and G12D). This study also investigated the effects of tipifarnib on the MAPK pathway. This was achieved by evaluating the pERK to ERK ratio through western blot (Figure 4C). After increasing dosages of tipifarnib, 293T cells displayed an increased ratio of activated pERK to ERK of nearly 300%. However, 786-0, A498, and Caki-2 cells showed a dose dependent decrease in pERK/ERK ratio with a depletion at the highest 0.5 uM tipifarnib doses of 61%, 25%, and 24%, respectively. 

### 3.6. Tipifarnib Suppresses Cell Proliferation in SS RCC Cell Lines

To investigate the effect of sunitinib and tipifarnib on tumor proliferation, an MTT assay (Figure 5A) was used. Cells were split into four groups: DMSO control, 2.5 µM sunitinib only, 0.5 µM tipifarnib only, and 2.5 µM sunitinib + 0.5 µM tipifarnib. On analysis of the MTT assay, no change in proliferation was seen from the 293T cells after treatments 0.5 µM tipifarnib only or 2.5 µM sunitinib only. However, after a dual-drug treatment, sunitinib and tipifarnib displayed a synergic ability to decrease 293T cell proliferation. Investigation of these trends in 786-O, A498, and Caki-2 cell lines revealed that A498 and Caki-2 showed no change in proliferation after sunitinib-only treatment, whereas 786-0 displayed a statistically significant decrease in proliferation (*p* < 0.05). Tipifarnib-only treatment showed the same pattern with A498 and Caki-2 cells, with no cytotoxic effect while causing a decreased proliferation of 786-O cells. After using a combination of 2.5 µM sunitinib and 0.5 µM tipifarnib (293T non-cytotoxic dose), a statistically increased cytotoxic effect was observed on all three cancerous cell lines. A combination index (CI) was calculated and demonstrated that these dual drug therapies were synergistic with one another.

This study subsequently correlated the MTT assay with cyclin D1 expression on western blot analysis after normalizing to actin (Figure 5B). After a 48-h incubation, significantly decreased cyclin D1 levels were observed in 293T cells for both sunitinib-only and tipifarnib-only groups (*p* < 0.05). After dual-drug treatment, 293T cells downwardly expressed cyclin D1 with an additive CI of 96.3. Cyclin D1 expression was also evaluated in the SS cell lines. No change in expression was found after tipifarnib-only treatment in these three SS cell lines. However, 2.5 µM sunitinib-only treatment caused all three cancerous cell lines to have decreased cyclin D1 expression when compared to DMSO control (*p* < 0.05). Lastly, after dual-drug therapy, 786-O, A498, and Caki-2 cells downwardly expressed cyclin D1. Cyclin D1 specific CI were calculated as the following: 786-O CI = 50.7% (synergistic), A498 CI = 51.8% (synergistic), and Caki-2 CI = 89.5% (synergistic). The MTT assay was used to determine the short-term (48 h) direct cytotoxic effect of sunitinib and tipifarnib, whereas the colony forming unit (CFU) assay was used to assess the long-term (two weeks) direct effect of this drug treatment (Figure 5C). The CFU assay confirmed the MTT assay, revealing that tipifarnib and sunitinib displayed a synergistic cytotoxic effect on both short- and long-term therapy in SS cancerous cells. 

### 3.7. In-Vitro Utilization of Tipifarnib for Sunitinib-Resistant RCC Cell Lines

To assess tipifarnib’s direct potential to combat tyrosine kinase inhibitor (TKI) resistance, the assays on SR cell lines were replicated. nSMase, Alix, and RAB27a expression was then evaluated on 786-O-SR, A498-SR, and Caki-2-SR cells after an increasing dose of tipifarnib (Figure 6A). nSMase, RAB27a, and Alix expression were found to decrease in a dose-dependent manner when compared to DMSO control. Activated pERK to ERK ratio was also assessed in the SR cell lines. After 48-h incubation with tipifarnib, pERK/ERK ratio showed a dose dependent reduction. More recently, Ma et al. reported that RCC cells induced PD-L1 expression via activation of the EGFR/ERK/c-Jun signaling pathway in RCC [37]. After treating three SR cells with a physiologically achievable dose of 0.5 µM tipifarnib for 48 h, exosome particle concentrations of all SR cell lines were reduced at *p* < 0.01 (Figure 6B). Furthermore, particle-size distribution ranged from 50 to 250 nm and was found to be comparable between treatment samples (Appendix A). As shown in Figure 4C and Figure 6A, pre-incubation of SS or SR RCC cells with tipifarnib caused substantial inhibition of ERK phosphorylation. Tipifarnib was observed as inhibiting PD-L1 protein expression more effectively in SR than SS RCC cells (Figure 6C) after testing the role of tipifarnib for PD-L1 expression in Caki-2-SS and Caki-2-SR cells. These data suggest that tipifarnib not only directly inhibited PD-L1 protein expression but also decreased exosome particle concentration in SS and SR RCC cells.

To assess the combination drug therapy of 2.5 µM sunitinib + 0.5 µM tipifarnib (Figure 6D), the three SR cell lines (786-O-SR, A498-SR, and Caki-2-SR) underwent an MTT assay. 786-O-SR, A498-SR, and Caki-2-SR showed no alteration in cell proliferation with sunitinib-only treatment (*p* > 0.05). Tipifarnib-only treatment displayed increased cell cytotoxicity on all three SR cells lines with *p* < 0.05. Combination therapy was then used to assess cell cytotoxicity. All three SR cell lines displayed a statistically significant reduction in proliferation after treatment with both tipifarnib and sunitinib (*p* < 0.05). SR cytotoxicity-specific CI were calculated as the following: 786-O-SR CI = 27.3 (synergistic), A498-SR CI = 21.1 (synergistic), and Caki-2-SR CI = 19.1 (synergistic). Finally, the MTT assay was correlated with cyclin D1 expression, using western blot (Figure 6E). Sunitinib-only treatment reduced cyclin D1 expression for A498-SR and Caki-2-SR while not affecting levels in 786-O-SR cells. Tipifarnib-only treatment reduced D1 expression in all three of SR cell lines (*p* < 0.05). Moreover, when employing combination therapy on all three SR lines, a downregulation of D1 expression was observed. SR cyclin D1 specific CI were calculated as the following: 786-O-SR CI = 90.3 (additive), A498-SR CI = 66.3 (synergistic), and Caki-2-SR CI = 77.6 (synergistic).

## 4. Discussion

Despite improvement in the early detection of renal malignancies, the overall mortality rate, particularly in patients with metastatic disease, remains stable. While small tumors can be successfully treated with partial nephrectomy, the ability to prolong survival in high-risk patients remains limited. Novel treatments and combination therapies continue to be evaluated and approved, but these treatments target the most common pathways of tumorigenesis, progression, and metastatic potential. However, bringing a drug from discovery to market can take more than 10 years and cost USD > 1 billion (about USD 3 per person in the US). Repurposing drugs reduces drug-discovery costs and shortens the availability timeline of new treatment options for patients. The accruing number of medications, which have passed the gambit of the FDA’s safety evaluation, gives scientists an opportunity to pinpoint previously undiscovered physiological targets useful in settings outside of the drugs-intended indication. By evading drug discovery, repurposing medications could yield new treatment regimens with the goal of improving outcomes without the exhaustive yet necessary four phases of clinical trials. Notably, this study demonstrated the effective repurposing of tipifarnib, a farnesyltransferase inhibitor, to combat RCC and the inevitable resistance to tyrosine kinase inhibitor therapy. 

Tipifarnib has recently passed phase II clinical trial for patients with mutated HRAS positive head and neck squamous cell carcinoma [38,39,40,41] and has been studied in numerous and disparate cancer types [40,42,43,44,45]. However, this study found that tipifarnib downregulated exosome production and secretion as well as PD-L1 protein expression in RCC. When used as an exosome-targeted therapy, tipifarnib directly attenuated exosome biogenesis by disrupting both the ESCRT-dependent and ESCRT-independent functional proteins Alix, nSMase, and Rab27a. This lab has previously highlighted these effects within prostate cancer cell lines [26]. Tipifarnib’s ability to reduce exosome concentrations was achieved at physiological-obtainable doses which were non-toxic to wild type 293T kidney cells. Additionally, tipifarnib was found to be an effective anti-cancer agent in both TKI (tyrosine kinase inhibitor) sunitinib-sensitive and sunitinib-resistant tumors through the mechanism highlighted in Figure 7. Of note, while the clinical data supporting higher expression of nSMase and RAB27a alone is correlated with lower overall survival on KM curve, increased Alix does not display this correlation. This discrepancy is possibly due to a change in Alix protein activity or concentration rather than mRNA expression. While this might be true, clinical data to support overall survival is correlated with increased Alix protein expression, activity, or decreased ubiquitination is unavailable. Additionally, alterations in Alix were mutually exclusive to a change in either nSMase, Rab27a, or both. When applying this to patient data, those with increased Alix and either mRNA expression of nSMase, RAB27a, or both, overall survival was significantly reduced. 

Advanced RCC treatment is still supportive in nature; however, advances have greatly improved life expectancy and quality of life. Sunitinib was first approved to treat advanced RCC in 2006 after showing an increased survival for patients [46]. While sunitinib has offered patients an effective treatment option, resistance has plagued outcomes. It has been uncovered that nearly 30% of patients are primarily resistant to TKI therapy before receiving their first dose. Additionally, the remaining 70% who respond positively, ultimately become resistant at a median of 14 months of therapy [47]. Qu et al. examined primary resistance in SR RCC and presented data to suggest its link to lncAR SR. When transcribed, this lncRNA produces miRNAs, shifting signaling pathways away from VEGF. Qu, et al. also examined secondary resistance [48]. This functional long-noncoding RNA, lncARSR, is identified as a part-exosome cargo and secreted into circulation for later absorption by SS RCC cells. IncARSR is then transcribed into miRNAs, conferring resistance where there previously were none. In this study, tipifarnib reduced the number of SR exosomes produced and secreted. When used in combination with sunitinib, tipifarnib treatment was able to overcome resistance and conferred tumor-specific death. 

Immune checkpoint inhibitors and PD-1 targeting antibodies, such as pembrolizumab, are currently a part of guideline-adherent care for the treatment of multiple cancer types including RCC [45,49]. This has been recommended because within the tumor microenvironment a major player in cancer immune evasion is mediated by the interaction surrounding PD-L1 [13,14]. However, immune escape has also been linked, in part, to the expression of the Fas/FasL pathway. This occurs through FasL expressed by RCC interacting with Fas and inducing apoptosis of T cells allowing for further immune escape [49,50]. Although FasL expression has a clear link to immune escape, the tumor mechanism to kill off T cells is not unimodal. Uzzo et al. investigated the Fas/FasL mechanism and functionally displayed the ability of some RCC cell lines to induce apoptosis in T cells in the setting of blocked FasL [51]. To highlight the importance of this study’s primary aim of reducing exosome burden, Yang et al. examined how RCC survives within the tumor microenvironment [52]. This examination showed that ACHN cell exosomes could trigger Jurkat T cell apoptosis, allowing RCC exosomes to evade the immune system by reducing interleukin-2 (IL-2), interferon-γ, IL-6, and IL-10 production by Jurkat T cells. Notably, this study expands upon the data of Yang et al. by examining the direct effect of SR exosomes on Jurkat T cells. First, after confirming Yang et al.’s findings in all three of these RCC cell lines, using SS exosomes, Jurkat T cells were treated with exosomes from SR RCC cells, a process which revealed an even greater level of Jurkat T cell-specific cytotoxicity compared to SS exosome treatment. This increased Jurkat T cell cytotoxicity, which was associated with an increased expression of PD-L1 on these SR exosomes, demonstrating the importance of exosome-targeted therapy in the setting of both TKI sensitivity and in patients who fail sunitinib treatment.

This study is not without limitations, the in vitro experimental design has been known to be an imprecise representation of the in vivo tumor microenvironment. In vivo cells likely experienced external stressors unaccounted for due to this study’s experimental design. Although this preclinical study establishes a clinical data-driven correlation between exosomal markers of biogenesis/secretion to both advanced tumor-staging and decreased overall survival, a genetically engineered mouse model is warranted. Furthermore, to confidently assess this combination therapy for clinical relevance, two separate phase IV clinical trials are needed on patients who presented for initial sunitinib treatment and on patients who failed TKI therapy. Phase IV trials will assist by evaluating if sunitinib + tipifarnib can truly either improve a patient’s life span, quality of life, disease-specific survival, or in combination. Additionally, the RAS mutation assay was employed to assess the presence of a mutation located at positions G12V and G12D. While these are commonly mutated sites within the RAS sequence, that does not exclude the possibility of a mutation occurring at another site such as Q61. Furthermore, while the target of this study was exosomes, showing modulation in exosome specific biogenesis and trafficking markers, it must be noted that ultracentrifugation does not wholly exclude the presence of microvesicles within our samples.

## 5. Conclusions

Repurposing drugs has a unique ability to reduce drug-discovery costs and improve treatment options for patients. This study showed that exosomes play a pivotal role in immunosuppression and contained higher PD-L1 expression than seen on the tumor cell surface. Using tipifarnib to inhibit exosome biogenesis and secretion through ESCRT-dependent and -independent pathways, this study demonstrated that exosome-directed therapy can be implemented with standard therapy to decrease tumor burden. Tipifarnib also showed synergic interactions with sunitinib to combat TKI resistance, thus having exciting potential as a new treatment option for patients with advanced RCC.

## Figures and Tables

**Figure 1 cancers-14-00903-f001:**
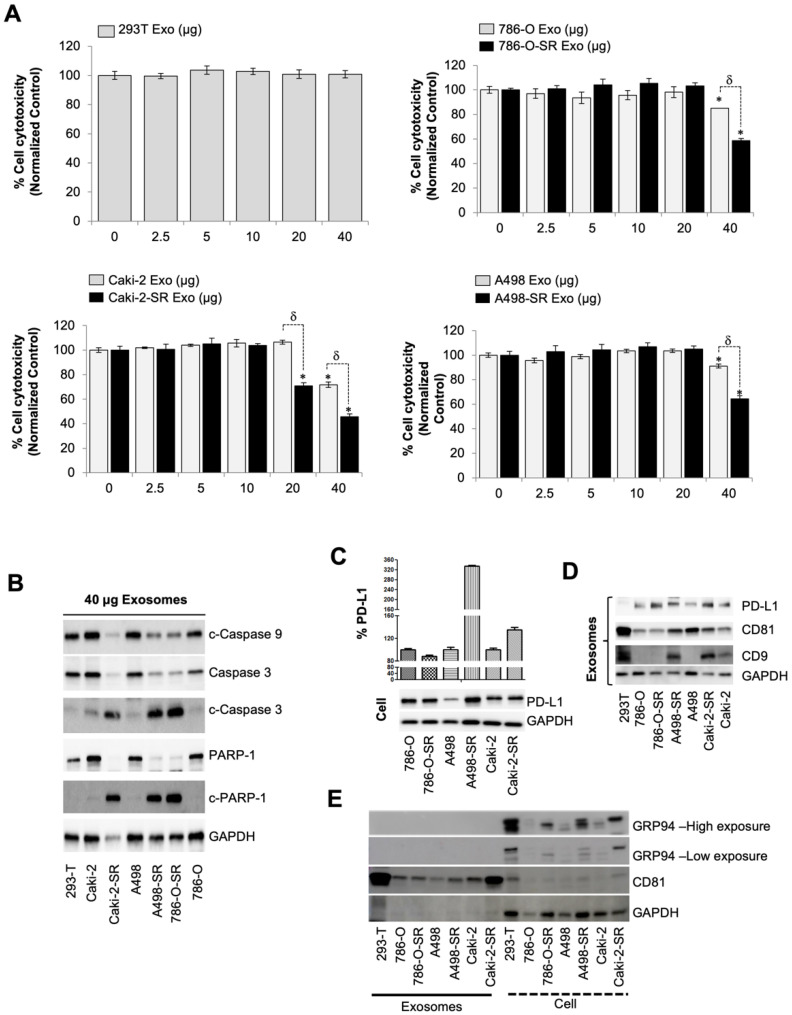
SR RCC exosome’s role in suppressing the immune system. (**A**) Effect of isolated exosomes from 293T or various RCC cells on Jurkat T cell proliferation determined by WST-8 assay after 72 h incubation (*n* = 4). (**B**) Caspase activation in Jurkat T cells after dosing with 40 µg/mL exosomes for 72 h. Caspases 3, 9, and PAPR-1 were then assessed using western blotting. GAPDH was used as an internal control. Full blots are presented in Appendix A. (**C**) PD-L1 protein in ccRCC cell lines was detected by western blotting. Full blots are presented in Appendix A. (**D**) Western blot analysis of the expression of exosomal PD-L1, CD9, CD81, and GAPDH in 293T, or SS, or SR RCC cell lines. (**E**) Western blot analysis of exosome and cell lysates identifying the presence of GRP94, a marker specific to large EVs, only in the latter. Full blots are presented in Appendix A. * *p* < 0.05, and ^δ^
*p* < 0.01.

**Figure 2 cancers-14-00903-f002:**
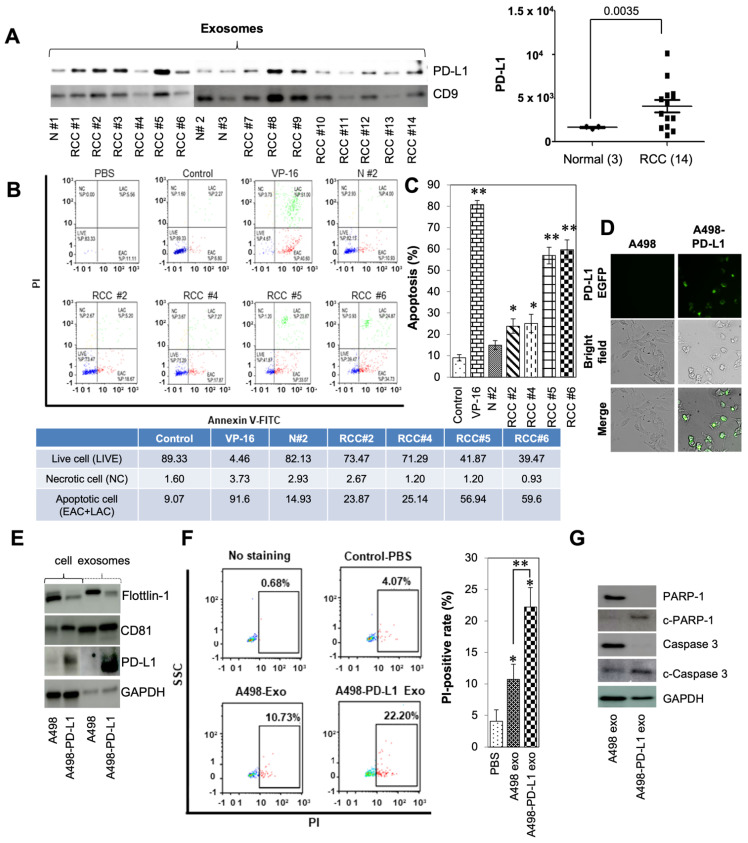
Effects of tumor-derived exosomes on human Jurkat T cells in patients with RCC. (**A**) Exosomes-enriched protein CD9 and protein PD-L1 were analyzed by western blotting among RCC patients and normal controls. (**B**) Apoptosis of Jurkat T cells after exposure to normal, RCC patient derived exosomes, or etoposide (VP-16) for 24 h. Apoptosis was analyzed by FACS using Annexin V-FITC/PI apoptosis kits. The induction of apoptosis was determined by flow cytometric analysis of PI-staining. Cells in the lower right quadrant (EAC; early apoptotic cells), upper right quadrant (LAC; late apoptotic cells), upper left quadrant (NC; necrotic cells), and lower left quadrant indicate live cells. (**C**) Representative results from three independent experiments. The data are presented as the mean ± SD. * *p* < 0.05, and ** *p* < 0.01 and compared with the normal group (N #2). (**D**) The A498 cells were transfected with PD-L1-EGFP vector. The cells were visualized under a fluorescent microscope after transfection. (**E**) Exosomes-enriched protein CD81 and protein PD-L1 were analyzed by western blotting. Full blots are presented in Appendix A. (**F**) Apoptosis of Jurkat T cells after exposure to A498, or PD-L1 overexpressed A498 derived exosomes, for 24 h. The induction of apoptosis was determined by flow cytometric analysis of PI-staining. Cells in the square box (apoptosis) indicated PI-positive. (**G**) A498 or A498-PD-L1 exosomes induced apoptosis activation in Jurkat T cells. Jurkat T cells were exposed to 40 µg/mL exosomes for 72 h, and the expression of caspases 3 and PAPR-1 were analyzed by western blotting. GAPDH was used as an internal control. Full blots are presented in Appendix A.

**Figure 3 cancers-14-00903-f003:**
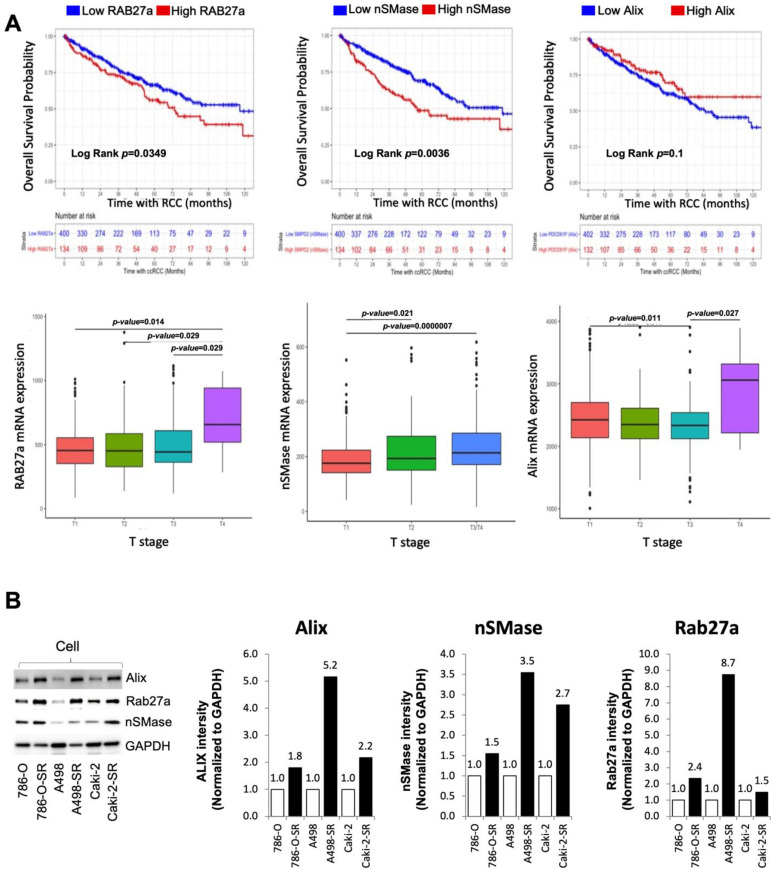
Clinical significance of Alix, nSMase, and RAB27a expression in RCC or SS and SR RCC cell lines. (**A**) Kaplan–Meier curve generated by implementing data from the TCGA database revealed that the high RAB27a expression group had significantly lower overall survival (OS) when compared to the outcomes of patients with low RAB27a mRNA expression. When comparing RAB27a mRNA expression between pathologic T stage, T4 displayed higher levels then all others. (**B**) Alix, nSMase, or RAB27a protein expression within each cell line were determined by western blotting. In all of the SR RCC cell lines, protein expression of Alix, nSMase, or RAB27a were higher when compared to each SS parent cell line. GAPDH was used as a loading control and full blots are presented in Appendix A.

**Figure 4 cancers-14-00903-f004:**
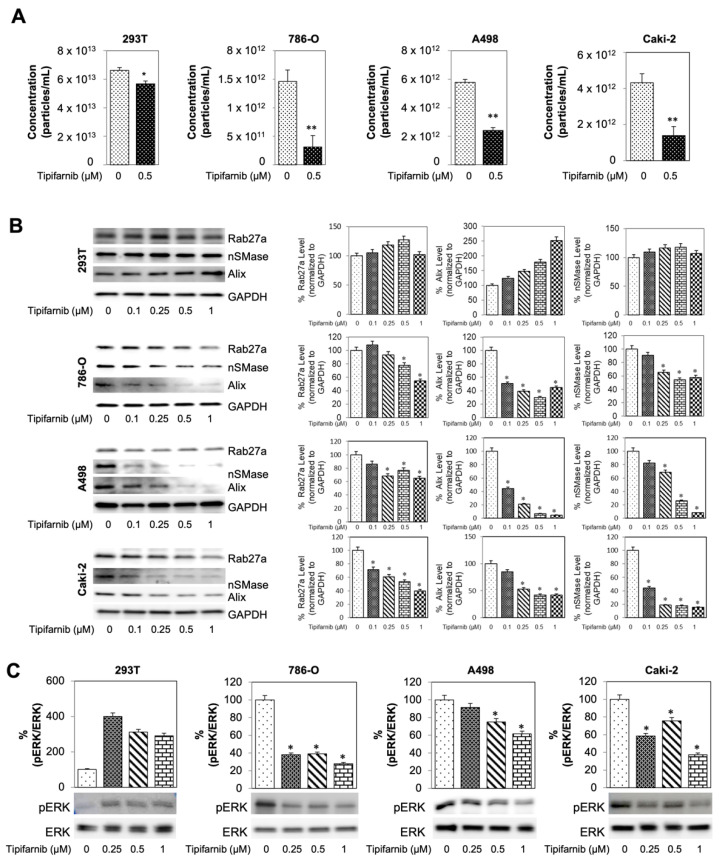
Tipifarnib reduces exosome load in sunitinib-sensitive RCC cell lines. (**A**) Measurement of exosome concentration across sensitive cell lines. After 0.5 uM of tipifarnib 786-O, A498, and Caki-2 displayed a decrease in overall exosome concentration. (**B**) Western blot analysis of exosome biogenesis and secretion markers Rab27a, nSMase, and Alix. All 3 markers display a dose dependent decrease in cancerous cell lines—A498, Caki-2, and 786-0. Either no change or dose dependent increase in these markers was seen in control 293T cells. (**C**) Activated ERK (pERK) to overall ERK ratio across sensitive cell lines. * *p* < 0.05, ** *p* < 0.01.

**Figure 5 cancers-14-00903-f005:**
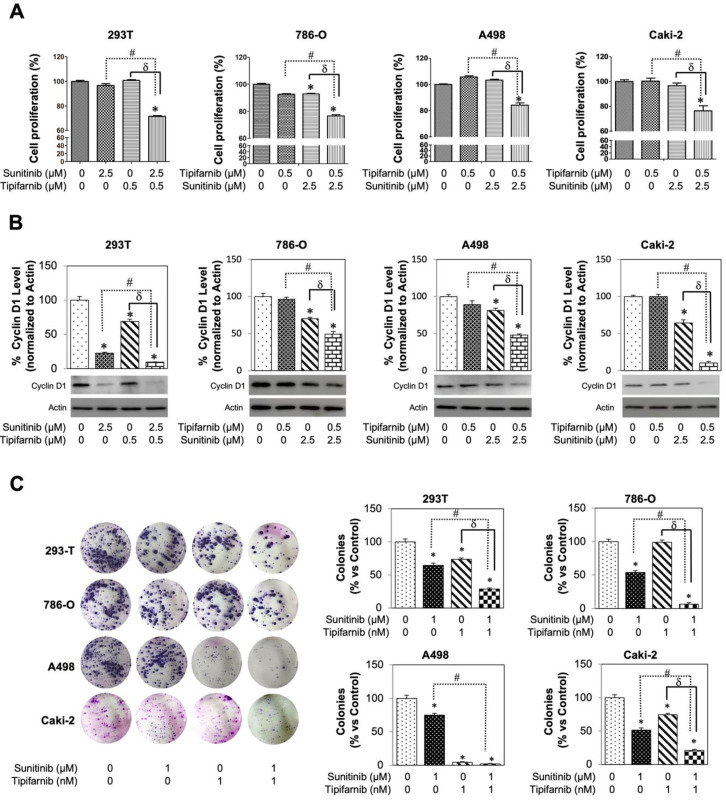
Sunitinib sensitive cells response to tipifarnib and combination treatment. (**A**) Sub-confluent 786-O, A498, and Caki-2 cells were grown in 96-well plates and dosed with sunitinib alone, tipifarnib alone, or combination of both sunitinib and tipifarnib. After 48 h, MTT assay was implemented to assess viability. All assays were completed in triplicates. * *p* < 0.05, compared with the DMSO controls. (**B**) Cyclin D1 levels after sunitinib-alone treatment decrease. These levels further decrease after combination treatment with tipifarnib and sunitinib. The densitometric plots show fold decrease in the expression. GAPDH immunoblot was used as a loading control for each western blot and full blots are presented in Appendix A. (**C**) Clonogenic assay was performed on all 3 SS cell lines and 293T set as our control. Colonies were counted 14 days after initial seeding. Each assay was completed in triplicate and repeated in at least three independent experiment batches. Representative results are shown, * *p* < 0.05, compared with the DMSO control groups, # *p* < 0.01, compared with the sunitinib groups, and δ *p* < 0.01, compared with the tipifarnib groups.

**Figure 6 cancers-14-00903-f006:**
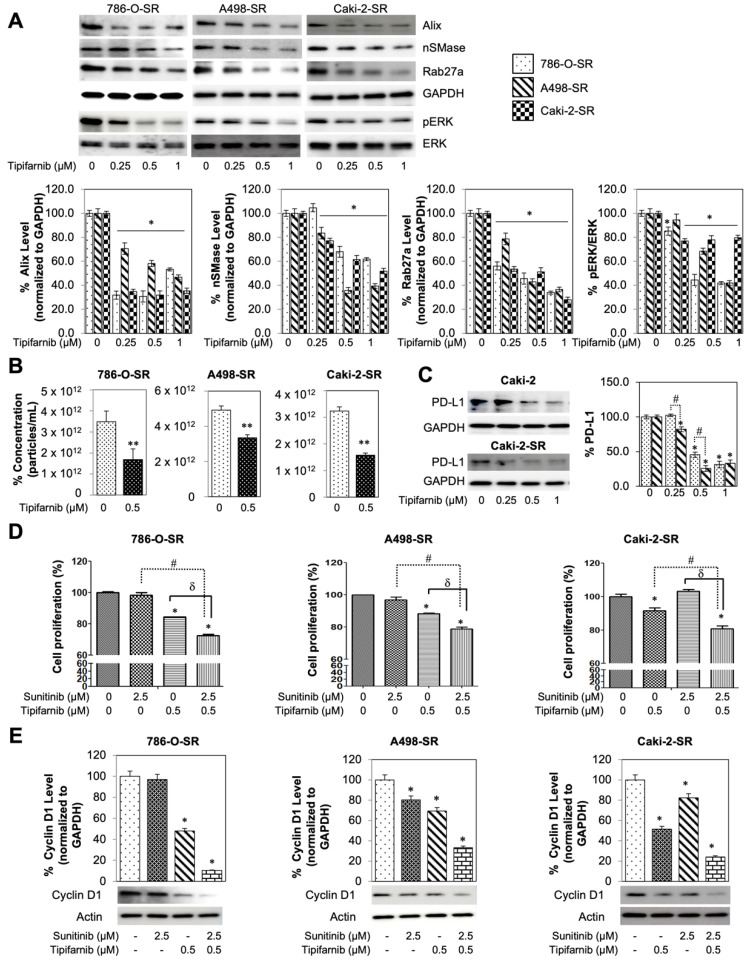
Sunitinib resistant cells response to tipifarnib and combination treatment. (**A**) Effect of tipifarnib co-exposure on SR RCC cells. Both ESCRT-dependent and -independent pathways are inhibited by tipifarnib but reduce the protein expression of Alix, nSMase, and Rab27a in a dose-dependent manner. Decrease in exosome specific biogenesis and secretion markers were seen for 786-O-SR, A498-SR, and Caki-2-SR cells. Full blots are presented in Appendix A. (**B**) qNano-IZON particle quantitative analysis (NP-100 nanopore) after tipifarnib treatment, depicting a significant decrease in exosome concentrations (50–250 nm size) in the CM of 786-O-SR, A498-SR, and Caki-2-SR cell lines. (**C**) PD-L1 levels decrease after tipifarnib treatment in both Caki-2 and Caki-2-SR cells. The densitometric plots show fold decrease in the expression of the signaling molecule. GAPDH immunoblot used as loading control was performed as previously described [21]. Full blots are presented in Appendix A. (**D**) Anti-proliferative implications of the dual drug therapy on WST-8 assay. (**E**) Sunitinib and tipifarnib’s effect on cyclin D1 levels, normalized to GAPDH. Each assay throughout this figure was done in triplicate and the results were repeated at least in three independent batches of experiments. Representative results are shown, * *p* < 0.05, compared with the DMSO control groups, ** *p* < 0.01, compared with the DMSO control groups, # *p* < 0.01, compared with the sunitinib groups, and δ *p* < 0.01, compared with the tipifarnib groups. Full blots are presented in Appendix A.

**Figure 7 cancers-14-00903-f007:**
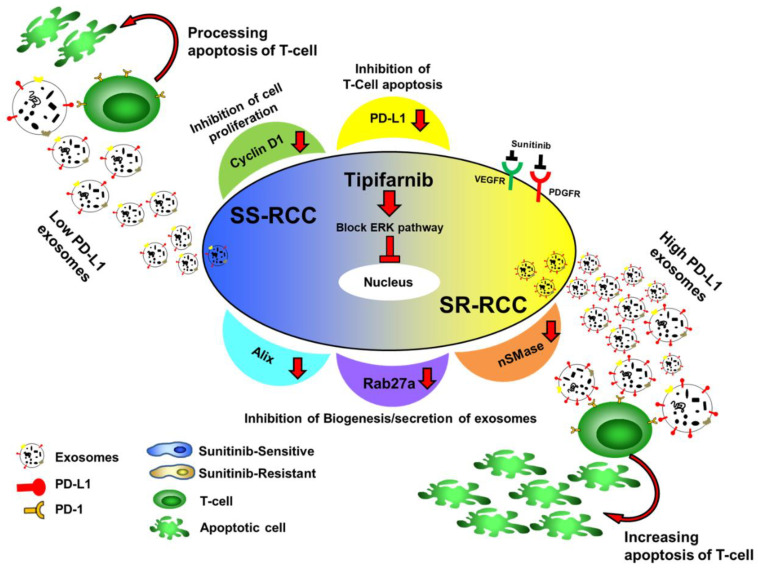
Illustrative diagram displaying this study’s proposed mechanism by which tipifarnib combats therapeutic resistance within the RCC microenvironment.

## Data Availability

The TCGA is a publicly archived dataset analyzed within this study. This database can be accessed by following the provided link: https://portal.gdc.cancer.gov/ (accessed on 25 January 2021).

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
