# Peer review of "Combination of Tipifarnib and Sunitinib Overcomes Renal Cell Carcinoma Resistance to Tyrosine Kinase Inhibitors via Tumor-Derived Exosome and T Cell Modulation"

_cancers, 2022, doi:10.3390/cancers14040903_

Round 1

Reviewer 1 Report

Thank you for the opportunity to review the manuscript entitled "Combination of Tipifarnib and Sunitinib Overcomes Renal Cell Carcinoma Resistance to Tyrosine Kinase Inhibitors via Exosome Reduction-Dependent Mechanism" submitted to Cancers by Dr. Krane and colleagues.  This is an important study, which has resulted in establishing tipifarnib as a potential treatment option for patients with sunitinib-resistant renal cell carcinoma, which is an area of significant unmet medical need.  The proposed mechanism by which tipifarnib functions in this system, through the inhibition/secretion of exosomes containing PDL1 is novel and important.  I have several minor stylistic comments and several more significant scientific questions/concerns, both of which are listed below:

1) In the abstract, the authors should mention the analysis on patient derived samples (Figure 3).  These important experiments boost the impact of the paper and highlighting them in the abstract may result in more readers for the paper.

2) ESRCT is used as an abbreviation in the abstract but is not defined

3) Sunitinib and tipifarnib should not be capitalized unless at the start of a sentence

4) I believe the authors are missing a larger point in the mechanism of action of tipifarnib.  This FTI is really specific for inhibiting the farnesylation and thus activation of HRAS as opposed to N and KRAS, as these other RAS isoforms have alternate mechanisms of prenylation.  This should be mentioned in the introduction/discussion.  In addition, the authors should include information about the HRAS mutation status of the cell lines used (HRAS is rare mutation in RCC).  Also, tipifarnib treatment results in an electrophoretic mobility shift of HRAS that can be observed by immunoblot and would be interesting to assess in conjunction with the pERK/total blots that the authors already show (eg Figure 4C)

5) Line 161 I believe the word "plasmid" is missing

6) Figure 2B and C - the authors might want to mention that the annexin positive cells quantified in panel C include both the PI positive and PI negative cells (upper and lower right quadrants).  Also by eye the % apoptotic cells for RCC#5 and #6 appear to be less than the mean shown in panel C, are there more "representative" FACS plots that could be included in panel 2B?

7) Figure 2E - could the authors explain the significance of tetraspanin and flotillin-1

8) The authors are truly to be commended for including the definition of what constitutes "high" expression for the K-M analysis in Figure 3A.  This is a critical detail that papers often fail to include.  However the labels in figure 3 are really very small and hard to see, if the font in this figure could be increased that would be most helpful

9) There appears to be some discrepancy between the clinical impact of Alix expression and its expression in the SR cell lines, could the authors comment?

10) Some additional references regarding tipifarnib in HRAS mutant malignancies to include:

Lee et al published a paper in Clinical Cancer Research describing a phase II trial of tipifarnib in patients with previously treated, metastatic, HRAS-driven urothelial carcinoma. 

Alonso-Alonso et al published a paper in Scientific Reports showing preclinical efficacy of tipifarnib in HRAS-driven models of T-cell acute lymphoblastic leukemia and T-cell lymphoma.

Hanna et al published a paper in Cancer describing an international phase II trial of tipifarnib in recurrent, metastatic, HRAS mutated salivary gland carcinoma. 

Gilardi et al published a paper in Molecular Cancer Therapeutics describing the preclinical efficacy of tipifarnib in models of HRAS-mutant head and neck squamous cell carcinomas.

11) Maybe highlight that Figure 5 is in SS cell lines to differentiate from Figure 6

12) With the decrease in cyclin D1 but no impact on cell viability at the 0.5 uM tipifarnib dose it might be of use to look at cell cycle analysis for the RCC cells with tipifarnib

13) The authors mention in the legend for Figure 7 that tipifarnib treatment leads to transcriptional downregulation of Alix, nSMase, Rab27a, however the authors have not examined the expression of these genes, they have looked at the protein expression.  It would be important to perform qPCR to make this claim as the differences in protein expression could be due to alterations in protein stability

14) The authors mention in the discussion that in vivo testing of the combination of tipifarnib and sunitinib would be needed, however, I don't think testing in cell line or patient derived xenografts as the author suggests would be helpful, as these experiments would need to be performed in immunodeficient mice, so the T cell apoptosis mechanism would not be able to be observed.  Testing the combination in a GEM model or xenografts established in a humanized mouse model would be necessary.

Author Response

Thank you for your comments. Please find the attached file with the authors responses. 

Reviewer 2 Report

Greenberg et al. described the study on combination therapy of tipifarnib + sunitinib that targeted exosome-conferred drug resistance using cell lines and patient-derived cells. There are concerns for this study listed as follows:

1)         Introduction

  1. The definitions of each EV, exosome and microvesicle need clarification. It is confusing that the authors use these terms alternately utilized without distinction. The small extracellular vesicles (sEVs or EVs) should be used instead of "exosomes" throughout this paper because the size distribution indicates the isolation of large particles. The ultracentrifugation isolates a mixed population of microvesicles and exosomes, in addition to other cellular particles.
  2. Line 73-75: Add references to the sentence in line73-75.
  3. Line 76-77: Briefly describe which part of exosome biogenesis is targeted by Tip.
  4. Line 91: difference in what?

2)         Methods: Line 125: Please explain FBS-exosome-free media. Is media free of FBS, or is media supplemented with exosome depleted FBS? If the latter, provide the source of this FBS.

3)         Results

  1. Figure 1A: % cell cytotoxicity on the y-axis implicates 100% cell death. If measured by WST-1, this could be cell proliferation. The analysis for cytotoxicity is different from the proliferation. Re-analysis is needed to claim cytotoxicity.
  2. Line 283-284: If there is a quantitative measure or the comparison made based on the quantitative analysis, please explain the detail of the comparison. Otherwise, the “higher PD-L1..." claim is misleading.
  3. Figure 3A: the figure resolution is too low and unable to read
  4. Figure 5C: Tipifarnib concentration unit is in nM, typo?
  5. All the EV western blots need negative cellular markers to show these EVs are free of cellular contaminants.
  6. EV weight (ug) standardization could influence the results as the ultracentrifugation will co-isolate non-EV-bound proteins. Standardization using particle count is recommended.

4)         Title: The phrase “via the exosome reduction-dependent mechanism” should be removed from the title, as there is no experimental evidence to show the dependence. The results show exosome biogenesis is possibly one of the downstream targets.  The phrase “partially through exosome-mediated regulation” suits better unless proving dependence with rescue experiments. As mentioned, the particle used in this study contains much more than exosomes that is difficult to specify the title’s claim.

Overall, it is difficult to understand the study due to poor organization and writing, which need improvement. However, it will become acceptable upon answering the above concerns and improving the language.

Author Response

(The authors gave the same response as above.)

Round 2

Reviewer 1 Report

I thank the authors for their kind words and for their careful attention to my previous concerns.  My concerns have been in large part addressed in the revised manuscript.  My one remaining concern is that the mutant RAS antibody sampler might not be sufficient to determine the RAS mutation status of the cell lines used, because HRAS is commonly mutated at Q61, which would not be detected by the antibodies in this sampler.  If the RAS mutation status of the cell lines is not available in the literature it might just be necessary to add that the cell lines could be mutated at Q61.  I would recommend targeted sequencing at the HRAS locus in these cell lines in the future. 

Author Response

Again, Reviewer #1 has offered us excellent insight into this project. Their comments are a clear example of the benefits the peer review process can yield. The authors agree with the above comment. While our assay tests for a common HRAS mutation site, that doesn’t mean a different mutated isoform doesn’t exist within our cells. We have added the below exert to our limitations paragraph in the discussion which can be found on lines 665-668. Future HRAS locus sequencing is underway in our lab.

“Additionally, the RAS mutation assay was employed to assess the presence of a mutation located at positions G12V and G12D. While these are commonly mutated sites within the RAS sequence, that doesn’t exclude the possibility of a mutation occurring at another site such as Q61.”

Reviewer 2 Report

The authors cleared most of the issues within the manuscript.

There are still a few concerning and misleading statements, thus it will become acceptable after responding to the following comments:

1)         Lines 82-92: The authors’ statement is inaccurate.

Despite that the size range of exosomes is smaller than that of microvesicles, their sizes largely overlap. Importantly, there is no method to separate these two. Ultracentrifugation (100,000xg) enriches an exosome population but cannot exclude microvesicles completely. The vesicle type is defined by the route of biogenesis: inward budding of the endosomal compartment form exosomes in the multivesicular bodies, MVB, which get released upon MVB-cell membrane fusion; microvesicles are the product of cell membrane budding. Please mention these facts in the manuscript.

Please refer to the following articles for more information.

https://doi.org/10.1080/20013078.2019.1648167

https://doi.org/10.1080/20013078.2018.1535750

The cup shape of the exosome is a form of artifact due to the drying process of TEM, and not the signature of exosomes.

https://dx.doi.org/10.3791/56482

2)         Methods: Line 125: Exosome-free media

Please use the term “EV-depleted media or exosome-depleted media”, as these are not “EV-free”.

Overnight ultracentrifugation cannot eliminate serum-derived EVs.

https://doi.org/10.1080/20013078.2017.1422674

3) Title: Please clarify the origin of the exosome in the title to avoid confusion.

Author Response

The authors would like to thank reviewer #2 for their time and effort. The author responses can be found in bold under each comment.

Reviewer #2

The authors cleared most of the issues within the manuscript.

There are still a few concerning and misleading statements, thus it will become acceptable after responding to the following comments:

1)         Lines 82-92: The authors’ statement is inaccurate.

Despite that the size range of exosomes is smaller than that of microvesicles, their sizes largely overlap. Importantly, there is no method to separate these two. Ultracentrifugation (100,000xg) enriches an exosome population but cannot exclude microvesicles completely. The vesicle type is defined by the route of biogenesis: inward budding of the endosomal compartment form exosomes in the multivesicular bodies, MVB, which get released upon MVB-cell membrane fusion; microvesicles are the product of cell membrane budding. Please mention these facts in the manuscript.

Thank you for your comment. The authors agree that the further clarification is needed on the definition of vesicle type and presence of other microvesicles within our samples. The two exerts have been added and can be found on lines 90-94 and 668-671, respectively.

“Exosome can be further differentiated from other EVs by the mechanism of biogenesis. Exosomes go through a process of inward budding from the cell surface to form the early endosome. These early endosomes later mature into multivesicular bodies which fuse with the cellular membrane allowing for the release of exosomes to the extracellular space. Microvesicles on the other hand are simply formed through membrane budding.”

“While the target of this study was exosomes, showing modulation in exosome specific biogenesis and trafficking markers, it must be noted that ultracentrifugation does not wholly exclude the presence of microvesicles within our samples.”

The cup shape of the exosome is a form of artifact due to the drying process of TEM, and not the signature of exosomes.

The authors are very sorry for the confusion. The mention of vesicle morphology has been removed from the manuscript and replaced with more clarification about the defining of exosomes from other EVs.

2)         Methods: Line 125: Exosome-free media

Please use the term “EV-depleted media or exosome-depleted media”, as these are not “EV-free”.

Overnight ultracentrifugation cannot eliminate serum-derived EVs.

The mention of “EV-free” has been replaced with “exosome-depleted media.” Thank you for working with us and homing in on the correct definitions. The authors appreciate your attention to detail by making this manuscript’s investigation clear to the reader.

3) Title: Please clarify the origin of the exosome in the title to avoid confusion.

Thank you for this comment. It is imperative we clarity the tumor specific origin of the exosome modulated by tipi in the title. Tipi has little to no effect on 293-T specific exosomes, which is a key distinction. The word “exosomes” in the title has been modified to reflect “Tumor-Derived Exosomes”.